# Impact of Oxidative DNA Damage and the Role of DNA Glycosylases in Neurological Dysfunction

**DOI:** 10.3390/ijms222312924

**Published:** 2021-11-29

**Authors:** Mirta Mittelstedt Leal de Sousa, Jing Ye, Luisa Luna, Gunn Hildrestrand, Karine Bjørås, Katja Scheffler, Magnar Bjørås

**Affiliations:** 1Department of Clinical and Molecular Medicine, Norwegian University of Science and Technology, 7028 Trondheim, Norway; jing.ye@ntnu.no (J.Y.); karine.bjoras@ntnu.no (K.B.); 2Department of Microbiology, Oslo University Hospital, University of Oslo, Rikshospitalet, 0424 Oslo, Norway; luisa.luna@rr-research.no (L.L.); Gunn.Hildrestrand@rr-research.no (G.H.); 3Department of Neurology, St. Olavs Hospital, 7006 Trondheim, Norway; katja.scheffler@ntnu.no; 4Department of Laboratory Medicine, St. Olavs Hospital, 7006 Trondheim, Norway; 5Department of Neuromedicine and Movement Science, Norwegian University of Science and Technology, 7491 Trondheim, Norway

**Keywords:** DNA glycosylases, oxidative DNA lesions, DNA repair, epigenetic markers, neurological disorders

## Abstract

The human brain requires a high rate of oxygen consumption to perform intense metabolic activities, accounting for 20% of total body oxygen consumption. This high oxygen uptake results in the generation of free radicals, including reactive oxygen species (ROS), which, at physiological levels, are beneficial to the proper functioning of fundamental cellular processes. At supraphysiological levels, however, ROS and associated lesions cause detrimental effects in brain cells, commonly observed in several neurodegenerative disorders. In this review, we focus on the impact of oxidative DNA base lesions and the role of DNA glycosylase enzymes repairing these lesions on brain function and disease. Furthermore, we discuss the role of DNA base oxidation as an epigenetic mechanism involved in brain diseases, as well as potential roles of DNA glycosylases in different epigenetic contexts. We provide a detailed overview of the impact of DNA glycosylases on brain metabolism, cognition, inflammation, tissue loss and regeneration, and age-related neurodegenerative diseases based on evidence collected from animal and human models lacking these enzymes, as well as post-mortem studies on patients with neurological disorders.

## 1. Roles, Sources, and Propagation of Oxidative Stress

Oxidative stress is an imbalance in the cellular levels of oxidizing and reducing agents (antioxidants). Oxidizing agents, such as reactive oxygen species (ROS), are formed during cellular metabolism or in response to inflammation. The brain has a high metabolic rate and accounts for up to 20% of the body’s energy demand. At the same time, it possesses low antioxidant capacity, which makes neurons susceptible to imbalanced ROS levels. Disrupted metabolism and subsequently increased ROS levels have been implicated in the pathology of several neurodegenerative diseases, such as Alzheimer’s disease (AD), Parkinson’s disease (PD), and Huntington’s disease (HD) (reviewed in [1]). A balanced ROS level is essential for brain homeostasis, as it provides an optimal redox state for the regulation of synapses and the formation of synaptic plasticity and memory (reviewed in [2]). An overview of ROS generation in the brain, the different oxidized products formed by ROS, and their role in triggering DNA glycosylase-dependent DNA repair or transcriptional regulation is illustrated in Figure 1.

The initial product of oxygen metabolism is superoxide (O_2_^•−^), which can be enzymatically converted to hydrogen peroxide (H_2_O_2_). Both molecules have low reactivity towards cellular components. However, in reaction with transition metals, H_2_O_2_ is reduced to hydroxy radical (^•^OH), a potent DNA-damaging agent, by the Fenton reaction. Environmental sources of ROS include ionizing radiation, which generates the hydroxy radical, and UVA radiation, which induces DNA damage via generation of the highly reactive singlet oxygen (^1^O_2_) [3,4]. The hydroxy radical and singlet oxygen can create various DNA-damaging events, including cyclopurine deoxynucleosides (cyPU) [5], apurinic/apyridimic (AP) sites, DNA single-strand breaks (SSBs), and base lesions (reviewed in [6]). Although the distribution and frequency of these lesions throughout the genome is unknown, it has been estimated that more than 70,000 DNA lesions are generated in a cell per day [7,8]. Importantly, the biological relevance of the detrimental ^•^OH radical to oxidative stress has been challenged. It has been shown that it is not a product of the Fenton reaction in the presence of bicarbonate, which is relevant for physiological conditions. Instead, the carbonate radical anion CO_3_^•−^ is the main ROS to be considered [9,10]. Whereas the ^•^OH radical is highly reactive, producing SSBs and base lesions, CO_3_^•−^ is more selective and serves as a one-electron oxidant that generates a guanine radical cation (G^+•^) and leads to the subsequent formation of 7,8-dihydro-8-oxoguanine (8-oxoG), as well as spiroiminodihydantoin (Sp) and guanidinehydantoin (Gh) lesions [11,12]. Guanine is the most sensitive base to oxidation due to its low reduction potential, and it has been hypothesized that CO_3_^•−^ reacts with DNA, forming an electron hole (h^+^) that can migrate along dsDNA and reside at low-energy sites susceptible to oxidation, such as sequences containing consecutive G tracks. Only G is oxidized to G^•+^ when the Fenton reaction occurs in an environment containing CO_3_^•−^, and G^•+^ subsequently generates 8-oxoG. Further 8-oxoG oxidation can form the hydantoin structures 5-carboxyamido-5-formamido-2-iminohydantoin (2Ih), Gh, and Sp [13]. Among oxidative DNA base modifications, 8-oxoG is considered one of the most abundant, generated by both ROS and reactive nitrogen species (reviewed in [14]). Besides the direct oxidation of guanine base in DNA, 8-OxoG can originate in the nucleotide pool as 7,8-dihydro-8-oxo-2′-deoxyguanosine 5′-triphosphate (8-oxo-dGTP), an oxidized form of dGTP.

Of particular interest are G4 motifs containing four or more oligo-G-sequences separated by seven or fewer nucleotides. These structures are enriched in telomers, promoters, and regulatory elements and are predicted to have regulatory roles during replication, transcription, translation, and telomere elongation (reviewed in [15]). G4 motifs form four-stranded G-quadruplex structures and are highly susceptible to oxidation. Interestingly, 8-oxoG sequencing revealed that this modification is enriched in promoters and UTRs in the mouse genome [16], as well as DNA replication origins in human and mouse cells [11]. Furthermore, accumulation of 8-oxoG in G4 structures has been shown to alter gene expression [17,18], and the presence of 8-oxoG, Gh, and Sp base lesions in G4 motifs has been shown to disrupt G-quadruplex formation at telomeres and to reduce telomere stability [19,20]. Further experiments are required to determine the interplay of different oxidative DNA modifications in promoter quadruplexes and their impact on transcriptional regulation. Furthermore, to understand the influence of oxidative stress on brain homeostasis, it is necessary to improve the accuracy of ROS-induced-lesion measurements and investigate which specific genomic regions are susceptible to oxidative stress in vivo.

## 2. Repair of Oxidative DNA Base Lesions by DNA Glycosylases

Base excision repair (BER) is the major pathway for repair of non-bulky oxidative DNA damage such as SSBs and base lesions, while cyPU lesions are exclusively repaired via nucleotide excision repair (NER) [21,22]. Nucleotide excision repair (NER) competes with BER in the repair of certain oxidatively generated guanine lesions (reviewed in [23]). The efficiency of NER and BER is determined by the type of lesion, type of DNA, and nuclear concentrations of lesion-recognizing enzymes within the two pathways.

BER is executed through several enzymatic steps, where a DNA glycosylase excises the damaged base in the first reaction (Figure 1, middle panel). There are six DNA glycosylases acting in response to oxidative base damage, namely 8-oxoguanine DNA glycosylase (*OGG1*), adenine-DNA glycosylase (*MUTYH*), endonuclease III-like protein 1 DNA glycosylase (*NTH1*/*NTHL1*), and Nei-like DNA glycosylases 1–3 (*NEIL1–3*). *MUTYH* is primarily a monofunctional DNA glycosylase that cleaves the N-glycosidic bond between the base and the deoxyribose. The phosphodiester bond at the remaining AP site is then cleaved and processed by AP-endonuclease 1 (*APE1*). Bifunctional DNA glycosylases, such as *OGG1*, *NTH1*, *NEIL1*, *NEIL2*, and *NEIL3*, possesses both the DNA glycosylase activity and an AP-lyase activity required to cleave the phosphodiester bond. The remaining steps include processing of the strand break by end-processing enzymes, repair synthesis by DNA polymerases, and ligation of the nick by DNA ligases (LIG) [24]. The resulting SSB can be processed by various end-processing enzymes to generate a free 3′-hydroxylgroup and 5′ phosphate. This is required prior to repair synthesis by DNA polymerases (POLs). The final repair of the gap can occur via short-patch repair orchestrated by the *XRCC1* scaffold protein involving single-nucleotide insertion by *POLβ* and ligation of the nick by *LIG1* or *LIG3*. The alternative pathway is long-patch repair in which the replication protein complex composed by *PCNA*, *FEN1*, *POLβ*, and *LIG1* extends, fills, and ligates the gap [24,25,26].

DNA glycosylases removing oxidized base lesions have broad and partly overlapping substrate specificities (Figure 1, middle panel). *OGG1* is the major enzyme for removal of 8-oxoG, and it also efficiently removes the ring-fragmented 2–6-diamino-4-hydroxy-5-formamidopyrimidine (FapyG) lesion from dsDNA only [27,28,29,30]. The *OGG1* enzyme does not excise oxidized bases from single-stranded DNA (ssDNA) or bubble (ssDNA flanked by duplex regions) structures. The *MUTYH* enzyme possesses adenine glycosylase activity on A:8oxo-G mispairs and prevents mutagenesis resulting from replication of unrepaired 8-oxoG [31,32,33]. Although not a DNA glycosylase, the mutT human homolog 1 (*MTH1*) enzyme provides an additional line of defence against the 8-oxoG in the genome. *MTH1* hydrolyses 8-oxo-dGTP to 8-oxo-dGMP in the nucleotide pool, thereby preventing misincorporation of oxidized nucleotides into DNA [34]. *NTH1* has a broad substrate specificity ranging from FapyG/FapyA to various oxidized pyrimidines [35,36,37]. These are also substrates of *NEIL1* and *NEIL2* [38,39,40]. However, while *NTH1* removes lesions from dsDNA, *NEIL1* excises from both ssDNA and dsDNA, while *NEIL2* prefers bubble ssDNA structures [41]. *NEIL1–3* DNA glycosylases remove oxidized purines Gh and Sp (hydantoins) from both ssDNA and dsDNA [42,43,44], where *NEIL3* appears to be the most efficient enzyme for removal of these lesions in ssDNA [44,45,46,47]. *NEIL1* and *NEIL3* can also remove Gh and Sp from telomeric and certain types of promoter quadruplexes [20,48]. The efficiency of removal is dependent on the type of G-quadruplex structure and the location of the hydantoin within the structure. Interestingly, G4 sequences are hotspots for hypoxia-induced oxidative DNA-base modifications, inducing the recruitment of BER proteins and accumulation of strand breaks in G4 promoter regions. In this scenario, *OGG1* and *APE1* recruitment to G4-containing promoter regions coincide with increased 8-oxoG and DNA strand-break levels in G4 sites in hypoxic rat pulmonary artery endothelial cells (PAECs) [49]. An unbiased genome-wide mapping of G4 structures, combined with mapping of AP site formation due to endogenous DNA-base oxidation and detection of binding of *APE1* and *OGG1* revealed a prevalence of AP sites in potential G-quadruplex−forming sequences in the genome of lung adenocarcinoma A549 and colon cancer HCT116 cells. *APE1* was shown to play a crucial role in regulating the dynamics of G-quadruplexes by promoting G4 folding via direct binding to G4 sequences [50]. The analysis of overlapping genes after enrichment via Chip-seq revealed co-occurrence of AP sites and G4 structures, as well as binding of acetylated *OGG1*, *APE1*, and acetylated *APE1* in promoter regions previously shown to form G4 structures. Based on these findings, in vitro data, and cell-based assays, it was proposed that *OGG1* is recruited to oxidized G-rich regions, where it removes 8-oxoG, generating an AP site, possibly destabilizing and opening the DNA duplex. *OGG1* remains bound to the AP site and is subsequently acetylated by histone acetyltransferase p300, coordinating the recruitment of *APE1*, which binds the AP site and stabilizes the G-quadruplex structure. Acetylation of *APE1* delays its dissociation from the AP site and promotes binding of transcription factors to these regions, leading to gene-expression activation or repression (Figure 1, right panel). Thus, in addition to maintaining genome integrity through the repair of oxidative DNA damage, DNA glycosylases play key roles in gene-expression regulation in specific sequence contexts.

## 3. Interplay between Epigenetics and BER

Epigenetics refers to mechanisms that regulate gene activity without altering the DNA sequence [51]. Methylation of cytosine bases in DNA is one such mechanism to carry epigenetic information via regulation of chromatin state. 5-methylcytosine (5mC) is induced de novo and maintained during DNA replication by DNA methyltransferases (DNMTs). Thus, chromatin states regulated by 5mC could be stable across cell divisions. However, 5mC is also a dynamic mark, which can be enzymatically modified in response to stimuli, particularly in gene-regulatory regions (reviewed in [52]). Ten–eleven translocation (TET) enzymes catalyze oxidation of 5mC to 5-hydroxymethylcytosine (5hmC) [53]. 5hmC is associated with active gene expression and is enriched in neurons and brain tissue, where the levels increase over time (reviewed in [54]).

Alterations in epigenetic modifications of cytosine impact proper regulation of gene expression in the brain and have been directly associated with neurological dysfunction. In the YAC128 (yeast artificial chromosome transgene with 128 CAG repeats) mouse model of Huntington’s disease, genome-wide loss of 5hmC was identified in the striatum and the cortex and correlated with abnormal neuronal function, development, differentiation, and survival [55]. Consistent with the importance of epigenetic modifications of cytosine in brain function, a marked decrease in global 5mC and 5hmC levels was reported in the hippocampus of AD patients [56]. Moreover, 5mC and 5hmC quantification in multiple brain regions (cerebellum, inferior parietal lobe, superior and middle temporal gyrus, and hippocampus/para-hippocampal gyrus) of patients in different stages of AD progression revealed already substantial alterations in these modifications in early stages of the disease, suggesting dysfunctional gene regulation via epigenetic mechanisms in AD progression [57]. Interestingly, Spruijt and colleagues discovered that *NEIL1* and *NEIL3* bind to 5hmC in mouse brain tissue [58], suggesting a non-canonical role of DNA glycosylases as readers of the 5hmC epigenetic mark.

5hmC is subjected to passive dilution during DNA replication, lacking a maintenance mechanism. However, in non-replicating cells, such as quiescent cells and postmitotic neurons, passive dilution is not achieved. In such scenarios, complete reversal of DNA methylation can be executed by iterative oxidation of 5hmC to 5-formylcytosine (5fC) and 5-carboxylcytosine (5caC) by TETs, followed by nucleotide replacement via BER initiated by the thymine DNA-glycosylase (*TDG*) [59,60,61]. *TDG* is plays an essential role in the development and maintenance of chromatin state during differentiation by recruiting chromatin-modifying enzymes and protecting CpG-rich promoter sequences from hypermethylation [62]. Interestingly, *NEIL1*, *2*, and *3* were able to rescue reactivation of gene expression from an in vitro methylated promoter after loss of *TDG* [63], suggesting that demethylation contributes to reactivation of gene expression and that NEILs are backup glycosylases in this process. In support of this, *NEIL1* and *NEIL2* accelerate 5faC and 5caC turnover by displacing *TDG* and promoting cleavage of the phosphodiester bond [64,65]. In contrast to previous observations, a direct interaction between *NEIL1* and *NEIL2* with oxidized methylcytosine residues was not detected, only in association with *TDG*. These results suggest that *NEIL1* and *NEIL2* may function as AP-lyases, rather than DNA glycosylases, during DNA demethylation in certain contexts (Figure 2).

*NEIL2* glycosylase activity has been shown to be stimulated by interaction with proteins in transcription-coupled repair [66]. However, as no change in baseline levels of oxidative damage was observed and NEIL enzymes facilitate *TDG*-induced DNA demethylation [64,65], it is possible that transcriptional differences in NEIL-deficient mice are caused by aberrant DNA demethylation rather than a DNA repair defect. Further experiments are required to resolve the specific functions of *NEIL1*/*NEIL2* in the context of various DNA structures, but also cell types, tissues, and organisms.

Interestingly, neural-crest abnormalities have been observed in *NEIL2*-deficient frog embryos [65], where *NEIL2* was shown to be involved in the removal of 5caC and 5fC. Knockdown of *TDG* and *TET3* also impaired neural-crest development [65], suggesting that reduced removal of 5caC and 5fC by *NEIL2*/*TDG* leads to this defect. However, inspection of mitochondrial DNA damage points instead towards a role of *NEIL1* and *NEIL2* in the protection of mitochondrial DNA and prevention of mitochondrial-induced apoptosis of neural-crest cells in frog embryos [67].

Reduced repair capacity of the oxidative base lesion Gh was observed in *NEIL3*^−/−^ neurospheres. Gh lesions are potential blocks to the replication machinery [68]; thus, a lack of proper Gh repair could potentially lead to a failure of proliferation and renewal of stem cells. Alternatively, dysregulation of DNA demethylation may be a potential mechanism for *NEIL3*-regulated neurogenesis. Altered genomic DNA methylation and gene expression have been observed in *NEIL3*^−/−^ mice, though in heart tissue [69]. Moreover, *NEIL3* binds to 5hmC in the brain [58] and rescues DNA demethylation in the absence of *TDG* [63].

Reinforcing a link between BER and epigenetic marks, SSBs were shown to accumulate within enhancers of post-mitotic neurons, supposedly due to BER processing of modified cytosine residues [70]. Although 8-oxoG has been among the most prevalent sources of DNA damage in the brain, this lesion did not preferentially accumulate in open chromatin or at enhancers and was therefore not assigned as a major contributor to the increased SSBs levels in neuronal enhancers. Conversely, mapping of oxidized forms of 5mC in the genome of post-mitotic neurons revealed that sites of DNA-repair synthesis and SSBs overlapped with 5hmC and 5fC peaks, and SSBs intensities correlated with 5hmC and 5fC intensities. These data indicate that cytosine methylation and demethylation are a potential source of site-specific DNA SSBs in human post-mitotic neurons, implicating a direct role of DNA SSB repair in hotspots of DNA damage as a fundamental process in neurological function, development, and ageing.

Importantly, base modifications beyond 5mC and its oxidized derivatives could potentially carry epigenetic information and regulate chromatin state and gene expression (reviewed in [71]). Indeed, 8-oxoG and its interaction with *OGG1* increased binding of the transcription factor NF-κB to its consensus sites at DNA [72,73,74]. Furthermore, the presence of 8-oxoG in potential G-quadruplex-forming structures (PQSs) on promoters can either increase or decrease gene expression, dependent on which strand contains the oxidated PQS [75,76,77].

Recently, it was demonstrated that knockout of histone deacetylase 1 (*HDAC1*) results in repression of genes critical to brain function. Promoters of the downregulated genes were enriched in G-rich sequences and increased 8-oxoG levels were detected in gene promoters from *HDAC1* knockout mice. It was shown that *HDAC1* deacetylates *OGG1*, which stimulates its activity. These results indicate that *OGG1* catalytic activity is responsible for 8-oxoG-mediated regulation of gene expression [78]. In agreement, generation of H_2_O_2_ during demethylation of histones by lysine-specific histone demethylase 1A (LSD1) results in a local increase in 8-oxoG, followed by recruitment of *OGG1* and topoisomerase IIβ. 8-oxoG-BER induces SSBs, which are utilized by topoisomerase IIβ to facilitate transcription initiation [79].

In contrast, a mechanism whereby catalytic activity of *OGG1* is not required for transcriptional regulation has also been proposed. In this context, an excision-deficient mutant of *OGG1* was found to be more potent as an activator of gene expression compared to a mutant with impaired substrate binding [80]. Additionally, increased recruitment of NF-κB, SP1, and TFID transcription factors was observed at the *CxCl-2* promoter upon binding of *OGG1* to 8-oxoG within the promoter, despite the fact that that 8-oxoG was not excised [74]. Reduced binding of these factors was demonstrated after treatment with an inhibitor that prevents *OGG1* from binding to 8-oxoG [81]. Moreover, enrichment of *OGG1* has been observed within introns upon oxidative stress, indicating a role of 8-oxoG as an epigenetic mark in gene-regulatory elements. In this context, *OGG1* was shown to be recruited to a DNA:RNA hybrid in intron 1 of the tissue inhibitor of metalloproteinase-1 (TIMP1) gene, leading to downregulation of TIMP1 in O3-exposed human airway epithelial cells and mouse lungs. A direct interaction between *OGG1* and an 8-oxoG-containing DNA:RNA hybrid, without excision of its substrate, was confirmed in vitro [82]. Hence, whether 8-oxoG-*OGG1*-mediated regulation of gene expression is dependent on *OGG1* catalytic activity may differ between cell types, tissues, and species.

## 4. Impact of DNA Glycosylases Removing Oxidative DNA Base Lesions on Cognitive Functions in Mice

*NEIL1*, *NEIL2*, *OGG1*, and *NTH1* display a widespread homogenously distributed expression in both human and rodent brain tissue, where *NEIL1* expression increases with age [83]. In contrast, *NEIL3* expression decreases with age and is restricted to the subventricular zone (SVZ) and the subgranular zone (SGZ) of the hippocampus, regions which are known to harbor neural stem and progenitor cells (NSPCs) in the mammalian brain [83,84]. A thorough characterization of the expression pattern of *MUTYH* in brain tissue has not been reported. Thus, despite the fact that most DNA glycosylase-deficient mice appear to have a normal lifespan, it has been of interest to investigate whether these mice differ in cognitive performance and behaviour compared to wild-type mice.

### 4.1. Distinct Roles of OGG1 and MUTYH in Cognition and Behaviour

To examine the impact of *OGG1* and *MUTYH* on learning and memory performance, single- and double-knockout mice have been monitored in the Morris water maze [85]. *MUTYH*^−/−^ mice displayed significantly improved memory compared to the other genotypes, while *OGG1*^−/−^ mice showed an opposite tendency. No clear difference in memory retention was found in *OGG1*^−/−^*MUTYH*^−/−^ mice, although they were significantly slower in learning the position of the platform compared to wild-type mice. In the same study, anxiety-like behaviour was examined in an elevated zero maze, where *OGG1*^−/−^*MUTYH*^−/−^ mice displayed reduced anxiety compared to wild-type and single knockouts. Notably, the strains had equal levels of genomic 8-oxoG in brain regions important for cognition and behaviour, suggesting that the impact of *OGG1* and *MUTYH* on cognitive performance is independent of 8-oxoG-BER. Interestingly, transcriptome analysis of *OGG1*^−/−^ and/or *MUTYH*^−/−^ mice revealed differentially expressed genes (DEGs) involved in estrogen receptor 1 (Esr1) signalling [85], which plays an essential role in the regulation of the memory-formation process [86,87]. Importantly, Esr1-induced gene expression is triggered by 8-oxoG-BER [79], raising the hypothesis that *OGG1* and/or *MUTYH* might regulate 8-oxoG levels at Esr1-activated gene loci. Recently, *OGG1*^−/+^ (heterozygote) mice were reported to have a transient reduction in learning performance, which was restored by exposure to low or moderate doses of X-rays [88]. A regulatory role in the striatum region of the brain has also been reported for *OGG1* [89]. Mice deficient in *OGG1* exhibited reduced motor function compared to wild-type mice on a rotarod test [90] and in an open-field test [89]. The impaired motor function was observed in aged mice (rotarod test: 5–6 months; open-field test: 26 months) but not in young ones (open-field test: 3 months), suggesting that *OGG1* maintains motor function in an age-dependent manner.

### 4.2. NEIL1 and NEIL2 Regulate Cognitive Processes in a Cooperative Manner

*NEIL1*^−/−^ mice displayed altered long-term spatial-memory performance in the water-maze test [91]. While wild-type mice (9–13 months) retained memory of the hidden platform for 5 days, the *NEIL1*^−/−^ mice remembered the platform location for only up to 24 h. Impaired hippocampal-dependent memory (both long-term and short-term memory) after *NEIL1* depletion has also been demonstrated in a novel object-recognition test, as well as a context fear conditional test [92,93]. Investigations in young male mice suggested that adult hippocampal neurogenesis was reduced in the absence of *NEIL1* [92]. Additional features of *NEIL1*^−/−^ mice include anxiety-like behavior, impaired olfaction, and increased neuroinflammatory response [93,94].

For *NEIL2*^−/−^ mice, on the other hand, no overt phenotype has been reported, but they have been shown to be susceptible to inflammation and to accumulate oxidative DNA damage in transcriptional regions [95]. In contrast, no global increase in DNA damage or mutation levels has been detected in mice lacking both *NEIL1* and *NEIL2* [96]. Interestingly, *NEIL1*^−/−^*NEIL2*^−/−^ mice were shown to have increased learning capacity in the Morris water maze [97]. Additionally, in an elevated zero maze and an open-field maze, *NEIL1*^−/−^*NEIL2*^−/−^ mice were more active when exploring the surroundings, and in the zero maze they spent more time in the open areas compared to wild-type mice and single knockouts, suggesting reduced anxiety in the *NEIL1*^−/−^*NEIL2*^−/−^ mice. *NEIL1*^−/−^ and *NEIL2*^−/−^ mice also spent more time in the open areas; however, the differences were modest compared to double knockouts. Importantly, transcriptome analysis of CA1 neurons in *NEIL1*^−/−^*NEIL2*^−/−^ mice revealed DEGs related to nuclear receptor signaling and synapse function. Most of the top upregulated genes overlapped with DEGs in *NEIL1*^−/−^ mice, while most of the top downregulated genes overlapped with DEGs in *NEIL2*^−/−^ mice. Together, these results suggest that *NEIL1* and *NEIL2* co-operate in transcriptional regulation of genes related to synaptic function, possibly for maintenance of hippocampal functions.

### 4.3. NEIL3 Modulates Spatial Learning and Memory

The significance of adult neurogenesis in humans is not yet clear, although accumulating evidence points towards important roles for neuroplasticity, cognitive functions, and behavior [98]. In rodents, the proliferation and differentiation of neural progenitor cells (NPCs) is essential for hippocampal plasticity and have an impact on learning and memory, spatial pattern separation, and possibly emotional control [99,100]. Neurospheres from adult *NEIL3*^−/−^ mice displayed reduced proliferation capacity [101]. While *NEIL3*^−/−^ mice appear phenotypically normal [46,102], they demonstrated an altered synaptic function of hippocampal neurons, as well as impaired learning and memory in a Morris water-maze test and reduced anxiety-like behavior in an elevated zero maze [101]. These results suggest that loss of *NEIL3* affects adult neurogenesis, resulting in reduced hippocampal plasticity. Although the mechanisms underlying neurogenesis maintenance mediated by *NEIL3* are not clear, dysfunction in Gh repair and DNA demethylation have been proposed to potentially contribute to *NEIL3*-regulated neurogenesis, as discussed in Section 3.

We have recently examined the impact of *NEIL3* deficiency in hippocampal CA1 neurons. Delayed CA1 maturation was observed in *NEIL3*^−/−^ mice, suggesting a role of *NEIL3* in neurodevelopment [103]. Hippocampal place cells encode spatial information in their firing patterns (place maps) that is correlated with the specific spatial position of the animal [104,105]. *NEIL3*^−/−^ CA1 place cells displayed normal spatial activity in an open-field environment and were able to change their firing patterns when entering a new environment. However, the spatial correlation for the same place-cell population in the familiar environment was significantly reduced when cells were recorded after one day, suggesting that *NEIL3*^−/−^ CA1 place cells were unable to retrieve the original place maps and exhibited impaired maintenance of encoded spatial information [103]. Comparison of the wild-type and *NEIL3*^−/−^ CA1 transcriptome during postnatal development and after spatial exploration revealed differential expression of genes essential for synaptic regulation, suggesting an important role of *NEIL3*-specific gene modulation in the functional development of the hippocampus and, thereafter, the hippocampal-dependent cognitive function in adults. In addition, the spatial-experience-induced activation of immediate early genes (IEGs), such as c-Fos and Arc, was significantly impaired in *NEIL3*^−/−^ CA1, implicating a role of *NEIL3* in the molecular correlates of memory engrams. Together, these results suggest that impaired learning, memory, and spatial performance in *NEIL3*^−/−^ mice could be explained by an altered hippocampal transcriptome, defects in neuronal functions, and experience-induced transcriptional responses. A summary of features observed in mice deficient in DNA glycosylases removing oxidized DNA bases and implications in cognitive functions and neurodegenerative disorders is provided in Table 1.

## 5. Impact of DNA Glycosylases on Acute Tissue Damage and Regeneration during Brain Injury

An ischemic stroke occurs when the blood supply to the tissue is blocked, resulting in an imbalance in metabolic supply and demand and subsequent tissue hypoxia. Restoration of the blood supply, i.e., reoxygenation, leads to excessive oxidative stress and tissue injury [120]. A variety of DNA lesions are induced upon cerebral ischemia, including 8-oxoG, AP-sites, and single- and double-strand breaks and may cause death of neurons and other cell types [121].

The hippocampus is one of the most sensitive regions to cerebral ischemic brain injury [122]. Ischemic events in the brain induce BER activation [123,124], and upregulation of *APE1* has been shown to protect hippocampal CA1 neurons in rats [125]. BER activity is related to the vulnerability of hippocampal cell types to stroke, as stroke-sensitive CA1 pyramidal cells in rat hippocampal slice cultures demonstrated lower BER capacity than stroke-resistant CA3 pyramidal cells [126]. Notably, gene polymorphisms in human *XRCC1* and *OGG1* have been associated with risk for ischemic stroke [127], suggesting that repair of oxidative damage may be important in the pathogenesis of strokes.

Stroke is a leading cause of death and long-term disability; however, there are currently no available therapies for regeneration of brain tissues. Interestingly, Ginsenoside Rd (GSRd), an ingredient of an herbal medicine with therapeutic effect on ischemic stroke in patients [128], induces early expression of *NEIL1* and late expression of *NEIL3* in rats, indicating that enhanced BER capacity is fundamental to the maintenance genomic integrity and improved outcomes following ischemic stroke [117]. Below, we discuss the impact of DNA glycosylases on recovery after ischemic stroke based on studies of DNA glycosylase-deficient mouse models.

### 5.1. OGG1 and NEIL1 Prevent Tissue Loss after Ischemic Stroke

Increased infarct area and neuronal cell death have been detected in *OGG1*^−/−^ mice 48 h after permanent middle cerebral artery occlusion (MCAO) [90], suggesting that *OGG1* protects against tissue loss under brain injury conditions. Under normal conditions, 8-oxoG, FapyG, and FapyA were not significantly altered in the cerebral cortex of *OGG1*^−/−^ mice. After occlusion, a marked increase in 8-oxoG, FapyA, and FapyG levels was detected in both the ipsilateral and the contralateral cortex of WT and *OGG1*-deficient mice, suggesting that these base lesions are stroke-induced. Notably, the increase in 8-oxoG and FapyG levels was more pronounced in the contralateral cortex of *OGG1*-deficient mice. Moreover, it was demonstrated that *OGG1*-deficient mice had reduced 8-oxoG incision activity both at basal conditions and after stroke. Functional redundancy between DNA glycosylases could possibly compensate for a lack of *OGG1* in the context of FapyG/FapyA excision, since these are substrates of *NEIL1*, *NEIL2*, and *NTH1* [37,38,39,40]. Furthermore, a robust increase in *OGG1* expression was detected in WT mice shortly after stroke, indicating that a rapid adaptive BER response is essential for neuronal protection. Notably, increased *OGG1* expression in the nucleus of primary cortical neurons subjected to oxidative stress coincides with dysfunction and structural damage to mitochondria, suggesting a selective protection of nuclear DNA by *OGG1*. These data indicate that dysfunction in mitochondrial DNA repair may be detrimental for neuronal viability after stroke.

Interestingly, trafficking of a TAT-modified form of the DNA glycosylase Endonuclease III (Nth) to mitochondria within 5 h after transient MCAO, followed by reperfusion, significantly reduced infarct volume in a dose-dependent manner [128]. Beneficial effects were not observed by trafficking a catalytically inactive *OGG1* enzyme or a mutant inactive Endonuclease III enzyme. These data reinforce the hypothesis that EndoIII promotes stroke attenuation via repair of oxidative lesions in mtDNA, supporting the use of mt-targeted DNA repair drugs to add tissue protection to reperfusion therapy.

The transcriptome of neonatal *OGG1*^−/−^/*MUTYH*^−/−^ mice was recently investigated after brain hypoxia reoxygenation, where altered gene expression patterns of inflammatory markers in the hippocampus/striatum were detected [110]. These results raise the hypothesis that *OGG1*/*MUTYH*-mediated transcriptional regulation is important for brain stroke outcomes.

*NEIL1*^−/−^ mice have increased susceptibility to ischemic brain strokes induced by MCAO [91]. Compared to wild-type mice, larger infarct volume was already detected in *NEIL1*^−/−^ mice 48 h after stroke. Loss of motor function was observed in both wild-type and *NEIL1*^−/−^ mice after MCAO. However, in contrast to wild-type mice, motor-function loss was persistent in *NEIL1*^−/−^ mice for 48 h. The affected brain regions had significantly higher levels of DNA strand breaks in *NEIL1*^−/−^ mice, suggesting that an increased number of cells underwent nuclear fragmentation and apoptosis in the absence of *NEIL1*. The levels of oxidative lesions in ischemic tissues were not reported for *NEIL1*^−/−^ mice [91]. However, incision activity of 5-hydroxyuracil was decreased in the ipsilateral area of *NEIL1*^−/−^ brains, whereas the contralateral area had similar repair activity as the wild type. These results could indicate higher levels of *NEIL1* substrates in ischemic tissue of *NEIL1*^−/−^ mice. Apparently, *NEIL2* deficiency does not influence ischemic-stroke outcomes in rats [117].

### 5.2. NEIL3 Promotes Regeneration of Damaged Tissue after Ischemic Brain Injury

After adult and perinatal brain injury, neurogenesis increases in the neurogenic niches of the brain, SVZ, and SGZ. New, immature neurons migrate from the SVZ to the site of injury, where they differentiate into neurons [129,130,131,132,133]. Thus, these niches where *NEIL3* is expressed [83] could be a potential source of neural regeneration. Occlusion of the common carotid artery performed in perinatal wild-type and *NEIL3*^−/−^ mice resulted in hypoxic cerebral ischemia in the ipsilateral hemisphere [46]. Tissue damage was observed in the cortex, hippocampus, striatum, and thalamus, whereas the contralateral hemisphere was indistinguishable from a sham-treated brain. Early after injury, no differences in tissue damage between genotypes were observed. Conversely, 42 days after hypoxic ischemia, *NEIL3*^−/−^ mice displayed a significantly smaller volume of reconstituted neuronal tissue compared to wild-type mice. Neurogenesis was induced in both genotypes upon injury, although the number of neural progenitors was lower in *NEIL3*^−/−^ mice.

Altogether, these studies suggest that both *OGG1* and *NEIL1* are important for neuroprotection upon brain ischemia, and *NEIL3* is involved in expansion of the stem-cell pool, which is essential for regeneration of damaged tissue. However, further experiments are required to determine whether oxidative base lesions accumulate and are the cause of increased cell death and poorer outcomes after stroke in the absence of DNA glycosylases. An overview of different roles played by DNA glycosylases upon acute injury induced by hypoxia, as well as their functions in different pathological stages involved in the progression of neurodegenerative disorders, discussed in the next section, is provided in Figure 3.

## 6. Oxidative DNA Base Lesions and DNA Glycosylases in Neurodegenerative Diseases

Neurodegenerative disorders cover a range of diseases characterized by progressive neuronal damage and cell death, with brain aging as a common determinant factor in the pathogeneses of these diseases (reviewed in [134]). Decline in cognitive abilities and behavioral deficits is strongly correlated with cumulative oxidative stress in aged animal models [135,136]. Furthermore, higher levels of oxidative species and related lesions, concomitant with decreased antioxidant capacity of the brain, are major factors contributing to the development of neurodegenerative disorders and dementia in aged humans [134,137]. As the blood–brain barrier protects brain cells from environmental genotoxic factors, it is believed that endogenous ROS are the primary source of oxidative DNA damage in the brain. The underlying pathology appears to be related to metabolic and mitochondrial dysfunction, resulting in increased ROS levels [138]. Additionally, increased levels of transition metals, such as iron, is a characteristic of neurodegenerative diseases, which generate ROS in contact with oxygen [139]. Iron and copper have been shown to inhibit *NEIL1* and *NEIL2* enzymes in cell extracts, suggesting that their accumulation both increase ROS and inhibit repair of ROS-induced damage [140]. In this section, we will discuss the impact of DNA glycosylases processing oxidative lesions in the pathogenesis of age-associated neurological disorders.

### 6.1. OGG1May Play a Protective Role against Alzheimer’s Disease

AD is the most common neurodegenerative disease characterized by a progressive loss of memory function, together with the histopathological hallmarks: extracellular amyloid-β plaques and intraneuronal neurofibrillary tangles. Various oxidized base lesions have been detected in AD brains [141,142,143,144]. Among those, 8-oxoG was shown to accumulate before the onset of dementia [145,146,147] and in regions associated with cognitive functions, such as the hippocampus in post-mortem AD brains [148], and in several mouse models for AD [148,149,150,151]. In line with increased 8-oxoG, reduced *OGG1* activity has been observed in patient brains at different disease stages, from mild cognitive impairment to late-stage AD [152]. Moreover, mutations in the *OGG1* gene, either resulting in a complete loss of *OGG1* activity or reduced repair capacity, have been reported to be specific to AD patients [153]. Gene polymorphisms in *OGG1*, but also *XRCC1*, have been associated with increased DNA damage in AD patients [154,155]. Interestingly, expression of BER genes encoding *APE1*, *POLβ*, *OGG1*, and *PARP1* was shown to be higher in brain tissue than in blood samples from AD patients, highlighting the importance of active BER in repairing antioxidant lesions in the brain. In blood samples, *OGG1* mRNA levels were lower in AD patients compared to healthy controls already in early disease stages and in patients with abnormal and normal cerebrospinal fluid levels of Aβ-42/Tau, suggesting that alteration in the *OGG1* mRNA profile may occur independent of plaque/Tau pathology and is an event preceding AD. Consequently, *OGG1* mRNA may be a candidate marker for deficient BER in individuals prone to develop AD. However, *OGG1* was not detectable at the protein level; therefore, whether the mRNA levels correspond to protein levels and enzyme activity remains to be elucidated [156].

In another study, expression levels of the BER genes *OGG1*, *APE1*, *MUTYH*, *NEIL1*, and *PARP1* were shown to be significantly lower in lymphocytes of AD patients, as compared to healthy individuals. In this study, expression of *XRCC1* was not altered between the groups, and no differences in methylation patterns in the promoter of these BER genes were found [157].

Very recently, it was shown that *OGG1* and *MTH1* prevented the progression of AD pathogenesis in an AD mouse model [113]. Mice lacking *MTH1* and *OGG1* displayed a marked increase in genomic 8-oxoG levels, slightly higher 8-oxoG levels in mitochondrial DNA, and increased intracellular Aβ/Tau accumulation. The high 8-oxoG content induced microglial activation and neuronal loss, resulting in compromised cognitive function at 4–5 months of age, indicating that loss of *OGG1* and *MTH1* accelerates AD pathology. Importantly, administration of minocycline, an anti-inflammatory antibiotic that inhibits microglial activation, substantially decreased the accumulation of 8-oxoG in the nucleus of microglia and inhibited microgliosis and neuronal loss. According to a gene-expression profiling analysis, *OGG1* and *MTH1* promote the expression of several neuroprotective genes that are critical to the suppression of AD pathogenesis. These findings suggest that efficient suppression of 8-oxoG accumulation in the brain could potentially be a novel approach for prevention and treatment of AD.

Furthermore, it was demonstrated that 38–52% of downregulated genes in an AD mouse model overlapped with downregulated genes in mice lacking *HDAC1*, which stimulates *OGG1* activity [78]. Interestingly, the promoters of these genes were enriched in G-rich elements. Loss of *HDAC1* in the AD mouse model exacerbated 8-oxoG in hippocampal neurons, and pharmacological activation of *HDAC1* reduced 8-oxoG levels and improved cognition in the AD mouse model, suggesting a potential therapeutic strategy for AD [78].

There are considerable associations between AD and BER; however, it is still not clear which roles the BER enzymes and DNA glycosylases play during the progression of the disease. Additionally, the contribution of mitochondrial DNA repair vs. nuclear DNA repair should be emphasized in future studies.

### 6.2. OGG1, MUTYH, and NEIL1 and Parkinson’s Disease

PD is primarily characterized by a loss of dopaminergic neurons in the substantia nigra (SN), leading to various motor impairments in PD patients [158]. Similarly, as for AD, the remaining neurons in the SN of PD patients harbor significantly more 8-oxoG, although in mitochondrial DNA. Notably, elevated levels of *OGG1* and *MUTYH* were also detected in mitochondria of PD patients (reviewed in [159]). Loss of *OGG1* in aged mice results in mild PD characteristics, such as decreased locomotor function, loss of neurons in SN, and decreased dopamine levels [89]. SNPs in *OGG1*, as well as *APE1*, have been associated with increased risk of PD upon exposure to pesticides [160]. Mice treated with the neurotoxins 6-hydroxydopamine (6-OHDA), 1-methyl-4-phenyl-1,2,3,6-tetrahydropyridine (MPTP), paraquat, and rotenone develop a PD-like phenotype, likely via alterations in oxidative metabolism and generation of oxidative stress [161,162]. Interestingly, *NEIL1*^−/−^ mice were shown to display accelerated motor dysfunction and neuroinflammation compared to wild-type mice upon treatment with 6-OHDA and MPTP, respectively [115]. These results suggest that *OGG1*, *MUTYH*, and *NEIL1* are associated with PD and may play distinct roles in the pathogenesis of the disease. Further experiments are required to determine the precise molecular functions of these DNA glycosylases in the development and progression of PD.

### 6.3. DNA Glycosylases Modulate the Pathogenesis of Huntington’s Disease in Mice

Several studies have suggested an impact of DNA glycosylases on HD pathogenesis. HD is a progressive neurodegenerative disorder characterized by striatal degeneration, cognitive deficits, and motor impairment. The disease is caused by expansion of CAG trinucleotide repeats (TNR) in the huntingtin gene (*HTT*). The resulting protein-containing long polyglutamine sequences interferes with cellular processes and causes neuronal defects and death [163]. In vitro excision of 8-oxoG by *OGG1* in a CAG template induced strand displacement, slippage, and formation of hairpins, leading to TNR expansion [164]. During BER of 8-oxoG in TNR, incorporation 8-oxodGMP from the dNTP pool and processing of the following 8-oxoG:A mismatch by *MUTYH* may also contribute to TNR expansion [165]. Importantly, error-free 8-oxoG-induced BER in TNR hairpin structures is dependent on strand-break processing by the endonucleases *MUS81*/*EME1* and *FEN1*, as well as repair synthesis by *POLβ* [166]. Loss of *OGG1* in mouse models carrying CAG repeats in the *HTT* gene resulted in delayed onset and decreased age-dependent expansion [164,167]. Similarly, a *NEIL1*-deficient HD mouse model displayed reduced somatic expansion of CAG repeats in brain regions [116,164]. Based on these results, it has been suggested that accumulation of base damage within CAG repeats induces an error-prone BER response that drives the TNR expansion in HD. Interestingly, treatment with a synthetic antioxidant inhibited trinucleotide expansion and rescued motor-function decline in an HD mouse model [167].

To determine the 8-oxoG-associated pathomechanisms of HD, mice lacking *OGG1*, *MTH1*, and/or *MUTYH* were challenged with the mitochondrial toxin 3-nitropropionic acid (3-NP), which recapitulates clinical hallmarks of HD [107]. Upon 3-NP exposure, *MTH1*/*OGG1* double-KO (DKO) mice displayed motor impairments and were shown to be highly susceptible to degeneration of medium spiny neurons (MSNs) in the striatum. The striatum is crucial for the coordination of motor function, cognition, motivation, and reward perception. These data suggest that *MTH1* and *OGG1* provide protection against oxidative stress in the striatum. Interestingly, 3-NP induced an increase in mitochondrial, but not nuclear, 8-oxoG levels in the *MTH1*/*OGG1* DKO striatum. Surprisingly, mice lacking *MUTYH* were resistant to the mitochondrial neurotoxicity and induced motor impairment induced by 3-NP, indicating that *MUTYH* enhances neurodegeneration. Loss of *MUTYH* activity in *MUTYH*/*OGG1* DKO mice resulted in less accumulation of both nuclear and mitochondrial ssDNA breaks, as compared to *OGG1*^−/−^ mice, suggesting that *MUTYH*-induced BER is responsible for the toxic effects of 8-oxoG. Furthermore, mice lacking *MUTYH* or both *MUTYH* and *OGG1* displayed similar resistance against 3-NP-induced motor dysfunctions. Altogether, these data demonstrate that *OGG1* and *MTH1* play protective roles against oxidative stress induced by 3-NP, while *MUTYH* mediates striatal degeneration upon 8-oxoG accumulation in the brain.

Microgliosis is associated with neurodegenerative disorders and activated microglia have been shown to mediate neuronal cell death [168,169]. The *MUTYH* protein was found to be expressed in microglia after 3-NP-induced striatal degeneration [107]. Additionally, in a mouse model of retinal degeneration, *MUTYH* enhanced microglia activation and promoted cell death upon oxidative damage, suggesting a pivotal role in neuroinflammation [170]. Although several studies report decreased inflammatory responses in *OGG1*-deficient mice [171,172,173], *OGG1* contribution to brain inflammation has not yet been systematically investigated.

### 6.4. OGG1/MUTYH, NEIL2 and NEIL3 Protect against Infectious Prion Disease

Prion diseases are genetic, sporadic, or infectious fatal neurodegenerative diseases, such as Creutzfelt-Jakob in humans, bovine spongiform encephalopathy in cattle, and scrapie in sheep. The pathogenesis of the disease is not fully understood but is characterized by misfolding and aggregation of the cellular prion protein (PrP^C^) into the proteinase-resistant PrP^Sc^ [174]. *OGG1*^−/−^
*MUTYH*^−/−^ mice inoculated with a prion strain exhibited reduced neuroprotection [111]. Although steady-state levels of oxidative base lesions were not measured, a shortened clinical phase in *OGG1*^−/−^
*MUTYH*^−/−^ mice correlated with reduced DNA integrity in the brain, suggesting that *OGG1*- and *MUTYH*-induced BER provides neuroprotection.

During development of prion disease, neural stem cells (NSCs) in the hippocampus accumulate and replicate prions that interfere with neuronal differentiation [175]. Thus, given its role in proliferation of NSCs and neurogenesis [83,84], the impact of *NEIL3* on prion disease has been investigated [119]. *NEIL3*^−/−^ mice inoculated with prions displayed a shorter clinical phase than wild-type mice, without other major alterations in disease characteristics. However, reduced expression of neural progenitor markers at end-stage supported a role of *NEIL3* in the activation of neurogenesis. Notably, no accumulation of the *NEIL3* substrate 5OHC was observed, and compensatory upregulation of redundant DNA glycosylases was not detected. Thus, repair of *NEIL3*-dependent oxidative lesions does not likely affect the clinical outcome of the disease, raising the possibility that *NEIL3* may exert a function beyond canonical BER in prion disease. Furthermore, recent results suggest that *NEIL2* also contributes to survival during prion disease in mice [118]. Impaired spleen function was observed upon loss of *NEIL2*, and transcriptome analysis revealed enhanced proliferation of immune cells and increased expression of mitochondrial genes at the end-stage of disease. Interestingly, significantly increased levels of 5mC were observed in *NEIL2*^−/−^ spleens, indicating an epigenetic mechanism mediating the pathogenesis of prion disease.

## 7. Conclusions

DNA glycosylases are important for brain function and protection against neurodegenerative disease, affecting neuroinflammation, cognitive processes, and neurogenesis. Accumulating evidence points towards a non-canonical role of glycosylases targeting oxidized DNA bases in transcriptional regulation and epigenetics. Because DNA glycosylase-deficient mice show no accumulation of mutations and minimal or no changes in DNA damage levels, it is likely that these enzymes may have specialized functions in tissues, possibly as regulators of DNA methylation and transcription. Further studies assessing catalytic activities of DNA glycosylases separately from substrate-binding properties would be valuable to determine whether the impact of DNA glycosylases on transcription, epigenetics, brain function, and disease rely on repair activity. Moreover, mapping of DNA glycosylase occupancy and distribution of oxidative lesions would aid in the elucidation of the functional roles of these enzymes in specific sequence contexts under normal or stress/disease conditions. It is possible that the mechanisms by which DNA glycosylases operate may vary between species, cell types, and genomic context. A deeper understanding of the contribution of DNA glycosylase activities the regulation of not only nuclear, but also mitochondrial DNA, can lead to the identification of selective therapeutic strategies to treat disorders associated with extensive ROS-mediated injuries.

## Figures and Tables

**Figure 1 ijms-22-12924-f001:**
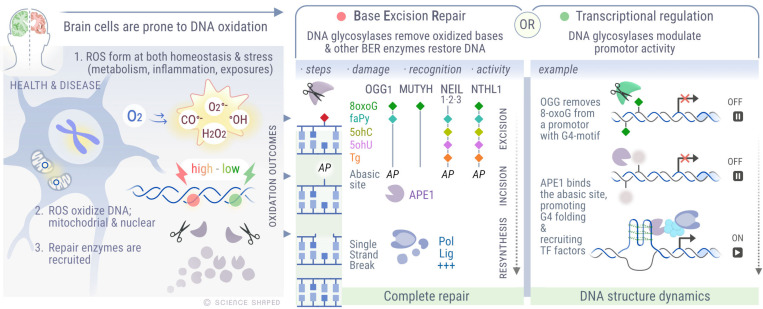
Outcomes of DNA oxidation in the brain mediated by endogenous levels of reactive oxygen species. ROS play key roles in brain homeostasis. However, at higher levels, ROS induce stress, culminating in the development of diseases (left panel). To counteract the harmful effects of ROS-induced DNA lesions, DNA glycosylases eliminate oxidized bases from DNA, restoring genomic integrity (middle panel). On the other hand, ROS-mediated base oxidation can also serve as a modulator of transcription in specific sequence contexts. For example, in G-forming quadruplex sequences, removal of 8-oxoG by *OGG1* leads to the recruitment of *APE1*, which promotes the folding of G4 structures in promoter regions and facilitates the loading of transcription factors, thereby modulating gene expression (right panel).

**Figure 2 ijms-22-12924-f002:**
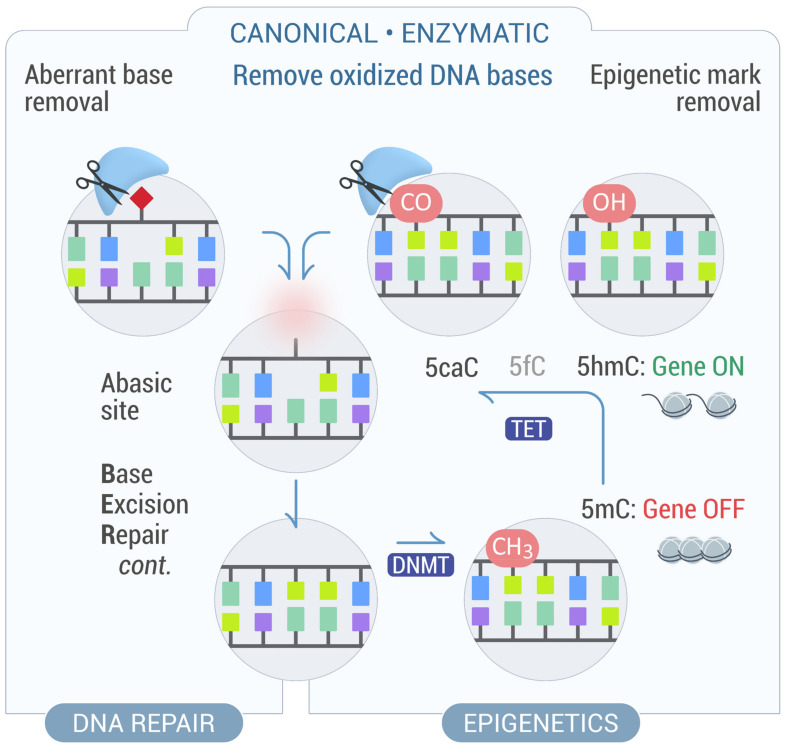
Canonical and non-canonical roles of DNA glycosylases in the removal of oxidative DNA-base modifications. DNA glycosylases can both remove oxidative base lesions from DNA, thus triggering BER, and remove specific epigenetic marks, thereby influencing gene expression.

**Figure 3 ijms-22-12924-f003:**
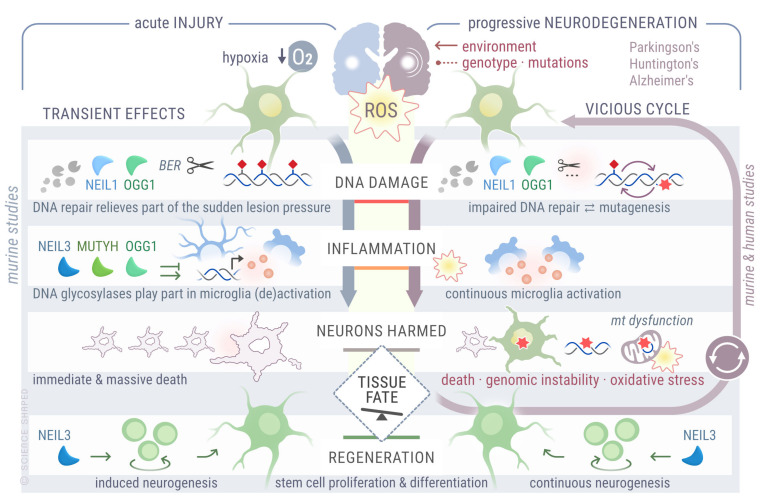
Overview of detrimental effects caused by increased oxidative stress upon hypoxia and during progressive degeneration, roles of DNA glycosylases in processing resulting oxidized products, and outcomes related to DNA damage repair, inflammation, and tissue regeneration.

**Table 1 ijms-22-12924-t001:** Features of mice lacking DNA glycosylase removing oxidative base lesions and the impact of DNA glycosylase loss in cognition and disease.

Genotype	Cognitive Function	Brain Ischemia	Neurodegenerative Disease	References
*OGG1* ^−/−^	Slightly reduced spatial memory; impaired motor function in aged mice	Increased brain-tissue damage	Mild PD clinical features; delayed onset and decreased expansion of TNR in HD	[85,89,90]
*MUTYH* ^−/−^	Improved spatial memory	N/A	Resistance to motor dysfunction in HD model	[85,106,107]
*OGG1* ^−/−^ *MUTYH* ^−/−^	Impaired learning; decreased anxiety	Altered expression of inflammatory markers	Resistance to motor dysfunction in HD model; reduced DNA integrity in prion disease	[85,107,108,109,110,111,112]
*OGG1* ^−/−^ *MTH1* ^−/−^	N/A	N/A	Accelerated AD pathogeneses; motor dysfunction and neurodegeneration in HD model	[107,113,114]
*NEIL1* ^−/−^	Impaired spatial and non-spatial memory; increased anxiety	Increased susceptibility to stroke; persistent loss of motor function	Accelerated motor dysfunction and neuroinflammation in PD model; reduced TNR expansion in HD model	[91,92,93,115,116]
*NEIL2* ^−/−^	Slightly reduced anxiety	N/A (no impact in rat)	Decrease survival of brain cells in prion disease	[95,97,117,118]
*NEIL3* ^−/−^	Impaired spatial learning and memory; reduced anxiety	Impaired regeneration of neuronal tissue in perinatal mice; low number of neural progenitors	Reduced expression of neural progenitor markers in prion disease	[46,101,102,119]
*NEIL1* ^−/−^ *NEIL2* ^−/−^	Increased spatial learning; reduced anxiety	N/A	N/A	[96,97]
*NEIL1* ^−/−^ *NEIL2* ^−/−^ *NEIL3* ^−/−^	N/A	N/A	N/A	[96]

N/A: Not available.

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
