# Peer review of "Impact of Oxidative DNA Damage and the Role of DNA Glycosylases in Neurological Dysfunction"

_ijms, 2021, doi:10.3390/ijms222312924_

Round 1

Reviewer 1 Report

The review article of Sousa et all covers two aspects: First, the molecular mechanisms of repairing oxidative damage to DNA in general and in the brain. Second, the impact of oxidative DNA base lesions and DNA glycosylases on cognitive functions and neurodegenerative diseases. The authors demonstrated a good knowledge of the field they decided to describe and are up-to-date with the literature. In my opinion, the manuscript is perfectly written, and I do not doubt that it may be of interest to the readers of IJMS. On this basis, I recommend accepting it for publication in its current form.

Author Response

We thank the reviewer for the comments and appreciate the reviewer's assessment.

Reviewer 2 Report

Comments and Suggestions for Authors
The review is very well written and readable, summarizing important information and referring the reader to original scientific articles considering more details, but I suggest integrating the introduction as follows:
In the introduction the authors should not forget to mention the 5 ', 8-cyclo-2'-deoxypurine specific tandem lesions exclusively formed by hydrogen atom abstraction from 2-deoxyribose residues induced by hydroxyl radical. It’s well known that NER is the only repair mechanism responsible for  removal  these DNA lesions, but their role in the neurodegeneration, and  their accumulation in the tissue specific manner in my opinion it cannot be indicated. I suggest to add this reference
REF: Philip J Brooks. Free Radic Biol Med . 2017 Jun;107:90-100. doi: 10.1016/j.freeradbiomed.2016.12.028. Epub 2016 Dec 21.

Lines 56-58: The hydroxy radical and singlet oxygen can create various DNA dam- 56 aging events, including apurinic/apyridimic (AP) sites, DNA single strand breaks (SSBs) 57 and base lesions (reviewed in [5]).
I suggest adding more to the Cadet  review (ref 5)  the recent publication of Chatgilialoglu et al
REF:  On the relevance of hydroxyl radical to purine DNA damage
Chatgilialoglu, C.; Ferreri, C.; Krokidis, M.G.; Masi, A.; Terzidis, M.A. Free Radic. Res. 2021, 55, 384–404.

Author Response

Our response:

We thank the reviewer for the fruitful comments and based on the reviewer’s assessment we have modified the manuscript to include cyPU lesions.

Lines 56-58: The hydroxy radical and singlet oxygen can create various DNA damaging events, including cyclopurine deoxynucleosides (cyPU) [5], apurinic/apyridimic (AP) sites, DNA single strand breaks (SSBs) and base lesions (reviewed in [6]).

Lines 96-97: Base excision repair (BER) is the major pathway for repair of oxidative DNA damage, both SSBs and base lesions, while cyPU lesions are exclusively repaired via nucleotide excision repair (NER).

References:

5: Chatgilialoglu,C. et al DOI: 10.1080/10715762.2021.1876855

21: Philip J Brooks. DOI:10.1016/j.freeradbiomed.2016.12.028

22: Krasikova, Y. DOI: 10.3390/ijms22126220